# Soyasapogenol C from Fermented Soybean (*Glycine Max*) Acting as a Novel AMPK/PPARα Dual Activator Ameliorates Hepatic Steatosis: A Novel SANDA Methodology

**DOI:** 10.3390/ijms23105468

**Published:** 2022-05-13

**Authors:** Radha Arulkumar, Hee Jin Jung, Sang Gyun Noh, Hae Young Chung

**Affiliations:** 1Interdisciplinary Research Program of Bioinformatics and Longevity Science, Pusan National University, Busan 46241, Korea; radhuspn@gmail.com (R.A.); rskrsk92@naver.com (S.G.N.); 2Department of Pharmacy, College of Pharmacy, Pusan National University, Busan 46241, Korea; hjjung2046@pusan.ac.kr

**Keywords:** fermented soybean, soyasapogenol C, SANDA methodology, AMP-activated protein kinase, peroxisome proliferator-activated receptors alpha, hepatic steatosis

## Abstract

(1) Background: Soyasapogenol C (SSC), a derivative of soyasapogenol B (SSB), is specifically found high in many fermented soybean (*Glycine max*) products, including Cheonggukjang (in Korean). However, the biological activities for preventing and treating hepatic steatosis, and the precise underlying mechanisms of SSC, remain to be explored. (2) Methods: A novel SANDA (structural screening, ADMET prediction, network pharmacology, docking validation, and activity evaluation) methodology was used to examine whether SSC exerts hepatoprotective effects in silico and in vitro. (3) Results: SSC had better ADMET characteristics and a higher binding affinity with predicted targets chosen from network pathway analysis than SSB. SSC induced the phosphorylation of AMP-activated protein kinase (AMPK) and stimulated the nuclear translocation of peroxisome proliferator-activated receptor alpha (PPARα), further enhancing PPAR response element (PPRE) binding activity in HepG2 cells. Concurrently, SSC significantly inhibited triglyceride accumulation, which was associated with the suppression of lipogenesis genes and the enhancement of fatty acid oxidation gene expression in HepG2 cells. (4) Conclusions: Soyasapogenol C, discovered using a novel SANDA methodology from fermented soybean, is a novel AMPK/PPARα dual activator that is effective against hepatic steatosis. Dietary supplementation with soyasapogenol C may prevent the development of hepatic steatosis and other diseases associated with fat accumulation in the liver.

## 1. Introduction

Dietary supplements are key resources that play a pivotal role in the food processing and pharmaceutical industries. The rate of discovery of novel, functional food additives from common dietary sources has dramatically increased in recent years [1]. Soyasaponins are oleanane-type triterpenoid glycosides that are mainly found in soybeans (*Glycine max*) [2]; they exhibit various biological activities, including anti-cancer, hepatoprotection, plasma cholesterol-lowering, and anti-viral activities [3,4,5]. Moreover, soyasaponins are metabolized by intestinal microflora to produce metabolites such as aglycones (soyasapogenols) in humans, which have been demonstrated to be more effective than other glycosides. Soyasapogenol A (SSA) and soyasapogenol B (SSB) exhibit various biological functions, such as anti-cancer, anti-inflammatory, and anti-neurodegenerative activities [6,7,8]. In addition, SSBs have limited bioavailability and a lower rate of absorption in the human gut [9]. SSC, which is not a natural aglycone of soyasaponins, is derived from SSB via acid hydrolysis [2,10]. A recent study suggested that SSC content was increased in fermented soybean products such as Cheonggukjang (CGJ), Doenjang, miso, Doubanjiang, and Tianmianjiang [11]. Thus, an investigation of the biological properties of SSC is essential to predict its potential beneficial effects over other aglycones. To date, there is no evidence regarding the hepatoprotective effects of SSC, and its underlying mechanisms have not been reported. 

Hepatic steatosis is characterized by the excessive accumulation of fat in the liver [12]. Hepatic steatosis is mainly caused by de novo lipogenesis and impaired fatty acid oxidation in the liver [13]. The accumulation of hepatic fat has been associated with metabolic disorders such as obesity and insulin resistance [14]. The energy sensor adenosine monophosphate (AMP)-activated protein kinase (AMPK) is involved in the control of cellular homeostasis, which can play a pivotal role in the regulation of glucose and lipid metabolism [15]. AMPK also regulates fatty acid synthesis by inhibiting sterol regulatory element-binding protein 1c (SREBP-1c), a crucial transcription factor, through X receptor (LXR) activity [16] and the inactivation of acetyl-CoA carboxylase (ACC) [17]. Moreover, small molecules can induce AMPK activation and inhibit hepatic steatosis and adipogenesis in 3T3-L1 adipocytes [18,19,20,21]. Berbamine, a natural bisbenzyl-isoquinoline alkaloid, attenuates hepatic steatosis by activating the SIRT1/LKB1/AMPK signaling axis in high-fat diet (HFD)-induced non-alcoholic fatty liver disease (NAFLD) rats. Therefore, compounds from dietary supplementation can activate AMPK, which can protect against hepatic steatosis. This therapeutic approach may be useful in the treatment of hepatic disorders associated with NAFLD.

Peroxisome proliferator-activated receptors (PPARs) belong to the nuclear receptor superfamily and are involved in various biological processes that are associated with metabolic syndrome, including dyslipidemia, insulin resistance, glucose and lipid metabolism, oxidative stress, and overall systematic energy homeostasis [22,23,24,25]. PPARα is mostly expressed in the liver, which controls the target genes that are involved in fatty acid metabolism [26,27]. It has also been demonstrated that PPARα-deficient mice have an impaired response to fasting and promote the development of fatty liver [28]. Interestingly, PPARα agonists also stimulate the phosphorylation of AMP-activated protein kinase and its downstream target ACC [29,30]. In addition, the natural compound magnolol counteracts hepatic steatosis via AMPK-dependent PPARα activation [31]. Thus, activating AMPK and PPARα may protect against hepatic steatosis and NAFLD development.

Hence, in the present study, a combination of structural screening, ADMET prediction, network pharmacology, docking validation, and activity evaluation (SANDA methodology) was used to screen potential hepatoprotective compounds. Among 15 distinct saponin compounds, SSB and SSC displayed the best binding affinity with target receptors. In addition, SSB and SSC both had drug-likeness properties; SSC had better ADMET characteristics and a higher binding affinity with predicted targets chosen from network pathway analysis than SSB. Furthermore, we evaluated SSC as a novel AMPK/PPARα dual activator and its ability to counteract hepatic steatosis in HepG2 cells.

## 2. Results

### 2.1. Screening of Effective Soyasaponins from Fermented Soybean (Glycine Max) Using in Silico Evaluation

A total of 15 soyasaponins from four distinct groups, such as group B soyasaponins, soyasaponin, 2,3-Dihydro-2,5-dihydroxy-6-methyl-4H-pyran-4-one (DDMP) saponin, and soyasaponin aglycones of fermented soybeans were obtained from previous reports (Appendix A) [11,32]. Among the 15 saponin compounds, soyasapogenol B (SSB) and soyasapogenol C (SSC) displayed the best binding affinity with metabolic syndrome-associated target receptors (Appendix A). Previous studies have suggested that SSB exerts various beneficial effects, such as anti-inflammatory effects and the inhibition of fat accumulation [33], while also having a low absorption ability in human intestines and therefore, low bioavailability [9]. Generally, similar chemical structures contribute to common pharmacological functions, and since SSC is a derivative of SSB, both were used for comparative computational studies and the further evaluation of hepatoprotective effects.

### 2.2. SSC Had Better Pharmacokinetic Properties Than SSB

Generally, drug-like ADME and pharmacological properties are important for the clinical application of candidates originating from natural products (61). Accordingly, we wanted to screen a potential compound with improved ADME properties and similar pharmacological functions. The SwissADME tool (http://www.swissadme.ch/, accessed on 26 October 2021) was used to calculate the drug-likeness based on Lipinski and Veber’s rules [34]. Appendix A shows that the drug likeness properties were satisfactory compared to SSB, and Figure 1 shows that the HIA, SP, Caco-2 permeability, and the BBB of SSB were 92.184936, −3.60213, 22.2528, and 6.35656, respectively. SSC improved HIA, SP, and Caco-2 permeability and BBB at 94.555879, −2.4077, 24.6316, and, 13.1666, indicating that it was a more effective compound than SSB. Cytochrome P450s are important enzymes for drug metabolism in the liver. The main subtypes of cytochrome P450 are CYP2D6, CYP2C9, CYP2C19, and CYP3A4. The results showed that SSC was predicted to be a CYP2C9 inhibitor, a CYP3A4 inhibitor, and a substrate for CYP3A4. These results suggested that SSC is metabolized in the liver. P-glycoprotein (P-GP) is a member of the ATP-binding transmembrane glycoprotein family (ATP-binding cassette (ABC)), which can excrete drugs or other exogenous chemicals from cells. The results suggest that SSC are all substrates of P-GP, and they may be actively exuded from cells via P-GP. SSC is predicted to be a P-GP inhibitor. Drug elimination is related to the molecular weight and hydrophilicity of the compounds. The prediction results show that the total SSC values were the highest. The results also suggest that the control compounds may be toxic in the AMES test; SSC and SSB compounds are not toxic and not carcinogenic in rats and mice. Thus, the predicted results indicate that the ADMET characteristics of SSC are higher than those of SSB. The predicted properties are listed in Table 1.

### 2.3. AMPK and PPARα Were Predicted as Targets of SSC Using Network Pharmacology 

With a probability of more than 0.1, a total of 209 targets of SSC and SSB were obtained from Swiss target prediction software; further, a compound–target network was constructed and visualized using Cytoscape. As shown in Figure 2A,B, the network was established, and it consisted of 154 nodes and 515 interactions. Notably, 55 overlapped targets were present in both SSB and SSC, including the PPAR genes. This indicated that SSC could achieve the same effects as SSB in similar ways. The protein–protein interaction network (PPI) was also constructed based on common targets (Figure 2C). The main targets of PPI networks were extracted by analyzing their degree and betweenness centrality. Proteins with a degree value greater than 2 were collected. These targets may be responsible for pharmacological action against hepatic steatosis.

The potential pharmacological mechanisms were predicted using Gene Ontology biological process (GO: BP), molecular function (GO: MF), cellular components (GO: CC), and the Kyoto Encyclopedia of Genes and Genomes (KEGG). By loading the data into the DAVID bioinformatics resource, 6.7 databases and 14 pathways were screened according to the KEGG analysis after filtering with the parameter of BH correction of the *p*-values to less than 0.05. As shown in Figure 2D, the results showed that the shared target proteins of SSB and SSC were mainly involved in GO: MF, which is associated with kinase activity and lipid binding, and GO: BP, which is associated with lipid metabolic processes, response to lipids, lipid biosynthetic processes, regulation of lipid metabolic processes, and lipid catabolic processes. Likewise, the KEGG pathway analysis revealed the involvement of target proteins in the AMPK signaling pathway, the PPARs signaling pathway, and so on. Thus, this signaling pathway may be regarded as the core mechanism of SSC against hepatic steatosis.

### 2.4. SSC Had a Higher Binding Affinity Than SSB with Target Receptors

Molecular docking is a versatile program for predicting the lowest energy conformation of a ligand molecule at the active site of the protein. Information on the nature of the interaction can also be obtained from the docking study. The molecular docking study of SSC with AMPK and PPARα was performed to predict the conformational structure of SSC with the highest affinity for AMPK and PPARα. The protein and ligand interactions are shown in Figure 3 and Figure 4, respectively. In the 3D structure of AMPK and PPARα, the docking simulation of binding between SSC had the best binding energy score compared to the control compounds AMPK: AICAR and PPARα: fenofibrate. SSC and AMPK were successful in producing the lowest binding energy score of −10.24 kcal/mol (Figure 3A), whereas PPARα with SSC produced a compound score of −8.95 kcal/mol (Figure 4A) for Auto Dock 4.2.6. One of the nine docked poses was selected as the best pose with the highest binding affinity among the different poses obtained from the docking results, which were clustered according to their binding affinity. The interaction of AMPK with SSC is due to van der Waals and hydrogen-bonding interactions, as observed from the 2D interaction diagram, and these were in good agreement with the results of the thermodynamic forces. SSC had a higher binding interaction for AMPK in at least two docking tool scores compared to the control and SSB (Appendix A). Likewise, SSC had a higher binding score for PPARα in at least two docking tool scores compared to SSB, which was almost similar to the control. Furthermore, SSC had a higher binding energy and the lowest intermolecular energy for AMPK when compared to the control, and a good score for PPARα (Appendix A). These results show that SSC had a higher binding interaction and a greater negative interaction energy, which makes it quite clear that SSC could be a better candidate than SSB. Therefore, an attempt was made to validate the computational hypothesis by linking it to in vitro evaluation.

### 2.5. Pharmacophore Validation of SSC and SSB Properties

The binding interactions of the most active docked conformation of the SSB, SSC, and the target proteins were identified using the Ligplot+ tool. All amino acids within the active site of the target protein were checked, and important binding interactions were identified. In addition to hydrogen bonding, the activity of the ligands and the receptors, including inhibition, was also influenced by electronic bonding, and hydrophobic and van der Waals interactions [35]. For the control simulation, acadesine was docked with AMPK. Acadesine activates AMPK and induces apoptosis in B-cell chronic lymphocytic leukemia cells, but not in T lymphocytes. Out of the four docking scores, the maximum best binding score possibilities were used for analysis. The binding energy of the AMPK control compound AICAR was −7.54 kcal/mol, SSB was −7.68 kcal/mol, and for SSC, it was −10.24 kcal/mol. The docking scores indicated that SSC binds more tightly to AMPK than to SSB and the control AICAR. Compared to the results of SSC, VAL 24, GLY 25, VAL 30, LYS 45, MET 93, GLU 100, GLU 143, ASN 144, LEU 146, ALA 156, and ASP 157 commonly shared the interactions shown in Appendix A. Likewise, SSC-PPARα bound to residues within PPARα and found several residues, including CYS 276, THR 279, LEU 321, MET 330, and VAL 332 (Appendix A). For the control simulation, fenofibrate was also docked with PPARα; Fenofibrate Intervention and Event Lowering in Diabetes (FIELD) and Action to Control Cardiovascular Risk in Diabetes (ACCORD); the activation of PPARα by fibrates has rarely reduced cardiovascular disease (CV) risk. The binding energy of fenofibrate was −9.67 kcal/mol, SSB was −5.82 kcal/mol, and SSC was −8.95 kcal/mol. The docking scores indicated that SSC binds more tightly to PPARα.

### 2.6. Molecular Dynamics (MD) Simulation 

The molecular dynamics simulation technique assists with mimicking the conformational changes of a protein–ligand system over time. The lowest energy binding poses of the SSB, SSC-AMPK, and PPARα complexes obtained from molecular docking were subjected to MD simulation, and the results were compared with the control simulation data. The structural stability evaluation of AMPK, SSC, and SSB with the control compound AICAR complex, and PPARα, SSC, and SSB with fenofibrate, were made from their RMSD plots (Appendix A, respectively). RMSD plots show that SSC had lower fluctuations when compared to the control and SSB within 5 nanoseconds. PPARα-SSC attains equilibrium more quickly compared to the control and SSB. The average RMSD values imply that there is less structural deviation of the SSC, resulting in an improved structural stability of the complex. The structural flexibility of AMPK and SSC with the control complex was assessed from their per-residue RMSF plot. Appendix A shows the RMSF plot of AMPK-AICAR-SSC-SSB, considering the Cα-atomic fluctuations. In the same way, Appendix A shows PPARα, SSC, and SSB in the control complex. The plot shows a similar structural stability for the PPARα-SSC complex with a low RMSF value, compared to the control, which suggests active binding of SSC with PPARα. The strength of the interaction between AMPK-SSC-SSB and PPARα-SSC-SSB can be measured in terms of its interaction energy. The validity of the molecular docking study was verified by measuring the interaction energies from the MD trajectories. The average interaction energy of the AMP-SSC and SSB complex was obtained by combining the contributions from the Coulombic and van der Waals energies. In Appendix A for AMPK-SSC-SSB-AICAR, and in Appendix A for PPARα-SSC-SSB-fenofibrate, the interaction energy value indicates the active interaction of AMPK-SSC and PPARα-SSC, resulting in the formation of a stable complex. These findings further validate the docking study.

### 2.7. SSC Inhibited Lipid Accumulation in Palmitate-Treated HepG2 Cells

The cytotoxicity of different concentrations of SSC (0–20 µM) in HepG2 cells was determined using an Ez-Cytox assay. No apparent cytotoxicity in cell viability was detected following treatment with SSC (20 µM) for 24 h (Figure 5A). Therefore, SSC was used in a range of non-cytotoxic concentrations (5, 10, and 20 µM) in subsequent experiments. To determine whether SSC inhibits lipid accumulation, palmitate-treated HepG2 cells were incubated in the absence or presence of various doses of SSC. Intracellular TG content showed that SSC significantly decreased palmitate-induced lipid accumulation (Figure 5B). In addition, lipid accumulation induced by palmitate was visualized using microscopic inspection following Oil Red O staining. As shown in Figure 5C, treatment with SSC resulted in the significant reduction of lipid accumulation in a dose-dependent manner in palmitate-treated HepG2 cells.

### 2.8. SSC Attenuated Hepatic Steatosis via Activation of AMPK in HepG2 Cells 

AMPK plays an important role in fatty acid metabolism, and it is partially associated with lipid accumulation and fatty acid oxidation genes [36]. To investigate the level of phosphorylated AMPK (Thr172) in SSC-treated HepG2 cells, Western blotting was performed. The effect of SSC on phosphorylated AMPK increased in a dose-dependent manner (Figure 6A) and significantly reversed the protein level of phosphorylated AMPK in palmitate-treated HepG2 cells (Figure 6B). Next, we evaluated the expression of genes involved in lipogenesis and fatty acid β-oxidation in palmitate-treated HepG2 cells. The mRNA expression of lipogenesis genes, such as ACC and FASN, was determined using RT-qPCR; SSC reversed the palmitate-treated increases in ACC and FASN levels (Figure 7D). Moreover, SSC significantly reversed the protein levels of PPARα and its target genes in ACOX-1 and CPT-1α palmitate-treated HepG2 cells (Figure 7A,D). Taken together, these results indicate that SSC acts as an AMPK activator and subsequently inhibits lipogenesis and stimulates fatty acid β-oxidation, thereby attenuating hepatic steatosis.

### 2.9. SSC Is an Activator of PPARα in HepG2 Cells

To confirm the activation of SSC on PPARα DNA binding activity, HepG2 cells were transiently transfected with the PPRE-3X-TK-Luc vector and PPARα expression vector, followed by treatment of the cells with different concentrations of SSC or 10 µM WY14643. Figure 7B shows that SSC treatment enhanced PPRE binding activity compared to that of the control. In particular, the binding activity of SSC (10 µM) was greater than that of the positive control, WY14643 (10 µM). Furthermore, SSC significantly increased the nuclear translocation of PPARα at the protein level compared to that of the positive control (Figure 7C). These results support the efficacy of SSC in PPARα activation. Taken together, our findings demonstrated that SSC could be a dual activator of AMPK/PPARα, thereby ameliorating hepatic steatosis in HepG2 cells.

## 3. Discussion

Numerous active compounds are present in fermented soybean products, which makes it difficult to select potential candidates. Evaluating the potential functional substances based on active compounds is still a challenge that is confronted by analysts because hundreds and thousands of conceivable compounds exist in soybean products. Hence, in the present study, an advantageous SANDA methodology introduced using a combination of structural screening, ADMET prediction, network pharmacology, docking validation, and activity evaluation was employed to screen biologically active substances in fermented soybean products, and to further explore the anti-hepatic steatosis effects. We demonstrated that SSC had better pharmacokinetics and biological properties than SSB, and evaluated its anti-hepatic steatosis effect on HepG2 cells.

Among the 15 soyasaponins, aglycone SSB and its derivative SSC showed lower docking scores for metabolic syndrome targets (Appendix A). These two compounds, SSB and SSC, had better drug-likeness properties and satisfied drug laws with no violations. Since SSB has a limited bioavailability and a lower absorption in the intestine [9], it is more efficient than SSB because it is produced by the acid hydrolysis of SSB [2,10]. Moreover, SSC levels appear to be higher in many fermented soybean products [11]. It has been reported to increase the aglycone isoflavone content and its bioavailability in soybean fermentation [37,38]. As expected, most of the ADMET properties of SCC showed a higher range and positive manner than SSB (Table 1). More specifically, SSC had better absorption and distribution factors when compared to SSB. Network analysis can be used to determine the molecular and cellular interactions of genes and proteins [39]. The PPI network-based approach suggests that the genes in response to lipid metabolism and metabolic syndrome are associated with disease within average betweenness centrality and the node degree. Furthermore, SSC showed a lower docking score and greater binding affinity for predicted potential targets, such as AMPK and PPARα, which were chosen from network pharmacology (Figure 2). Based on the in silico results, the effects of SSC on AMPK and PPARα activation and anti-hepatic steatosis in HepG2 cells were investigated.

Hepatic steatosis is the first stage in the development of NAFLD, and it has been associated with various metabolic disorders, including obesity and insulin resistance [13,14]. Thus, discovering effective compounds from regular diets that inhibit lipid accumulation and enhance fatty acid oxidation is vital for the prevention of NAFLD. AMPK is an energy sensor that regulates cellular hemostasis, glucose, and lipid metabolism [15]. In the present study, SSC activated phosphorylated AMPK levels in HepG2 cells in a dose-dependent manner (Figure 6A). Furthermore, intracellular TG content and lipid accumulation were significantly decreased by SSC in a dose-dependent manner in palmitate-treated HepG2 cells (Figure 5). Since AMPK activation can inhibit hepatic lipogenesis and trigger fatty acid oxidation, we evaluated the expression of genes involved in lipogenesis and fatty acid β-oxidation in palmitate-treated HepG2 cells. The mRNA expression of ACC and FASN was significantly decreased by SSC in palmitate-treated HepG2 cells; meanwhile, ACOX-1 and CPT-1α expression was significantly increased. Similar to our results, WS070117, a novel AMPK activator, is involved in anti-hyperlipidemia and anti-steatosis effects [19]. Moreover, several studies have demonstrated that natural compounds ameliorate hepatic steatosis by activating AMPK-dependent pathways [31,40,41]. Taken together, these results revealed that SSC could be an excellent candidate for the prevention of hepatic steatosis through the activation of AMPK.

PPARα, a nuclear receptor, regulates genes that participate in fatty acid oxidation and impedes lipid deposition in the liver by promoting fatty acid catabolism [26,27,42,43]. Moreover, CPT1 and ACOX1, the enzymes involved in fatty acid oxidation, are important target genes of PPARα that can enhance mitochondrial activity [44]. In the present study, SSC increased the nuclear translocation of PPARα in HepG2 cells (Figure 7A). In addition, SSC treatment enhanced PPRE binding activity, and the binding activity of SSC was greater than that of the positive control, WY14643 (PPARα agonist) (Figure 7B). These results support the efficacy of SSC in PPARα activation, and it could be a PPARα agonist. Furthermore, we confirmed earlier that ACOX-1 and CPT-1α expression was significantly increased by SSC in palmitate-treated HepG2 cells, indicating that SCC promoted fatty acid oxidation. A recent study demonstrated that catalpol attenuated hepatic steatosis by activating PPARα-mediated fatty acid β-oxidation [45]. The findings of our study demonstrate that SSC enhances fatty acid oxidation by activating PPARα-mediated gene transcription and increases CPT1 and ACOX1 activity, thereby counteracting hepatic steatosis.

## 4. Materials and Methods

### 4.1. Compound Screening

Fermented soy foods contain many transformed bioactive compounds, including soyasaponins [3,4,5]. In total, 15 soyasaponins from CGJ were retrieved from previous reports (Appendix A) [11,32]. Based on the structural similarity evaluation, molecular docking of SSB and its derivative, SSC, were used for further experiments.

### 4.2. Pharmacokinetic Properties Prediction

ADMET prediction is critically important for drug discovery and the prediction of the accuracy of drug candidates. In silico tools have revolutionized disease management due to the early prediction of the absorption, distribution, metabolism, excretion, and the toxicity (ADMET) profiles of chemically designed and eco-friendly next-generation drugs [46]. To examine possible drug-likeness properties, molecular weight (MW), hydrogen bond acceptors (HBA), the number of hydrogen bond donors (HBD), the number of rotatable bonds (RB), and the lipophilicity (LogP) for selected compounds were obtained via the SwissADME tool (http://www.swissadme.ch/, accessed on 26 October 2021) [34]. The ability to satisfy most rules of Lipinski’s rule of five [47] and Veber’s rule of 3 [48] suggests the worthiness of being orally active drug candidates. PreADMET (https://preadmet.bmdrc.kr/adme/, accessed on 22 July 2021) [49] online software was used to estimate the pharmacokinetic properties of selectively screened compounds, and the absorption of drugs depends on factors including human intestinal absorption (HIA), membrane permeability (as seen in colon cancer cell line Caco-2), MDCK cell permeability, and skin permeability levels (SP). The distribution of drugs depends on factors such as plasma protein binding and the blood-brain barrier (logBB), which are predicted based on the CYP450 models for inhibition or substrate (CYP2C19, CYP2C9, CYP2D6, and CYP3A4). Excretion was predicted based on the total clearance model of Pgp inhibition, and the toxicity was predicted based on AMES toxicity, carcino mice, and carcino rats, and these parameters were calculated and checked for compliance with their standard ranges.

### 4.3. Target Prediction and Network Pharmacology

The computational tool for predicting targets, Swiss Target Prediction (http://www.swisstargetprediction.ch, accessed on 17 December 2021), was used to retrieve potential targets from homo species [50]. Protein–protein interactions (PPIs) with a confidence score range >0.5, were obtained after omitting duplicates using the STRNG database V.11.0, (https://string-db.org/, accessed on 28 December 2021) [51]. PPIs and predicted-target networks were constructed and viewed using Cytoscape software V.3.2.1 [52], with the analyzing network being the default setting with a “degree” value, and the pathway enrichment analysis (KEGG analysis) for all proteins/genes were subjected to DAVID Bioinformatics resources 6.7 databases [53]. Enrichment analysis was performed using G: profiler [54].

### 4.4. In Silico Experiment Materials

Three-dimensional chemical structures of SSB and SSC, along with the crystalline forms of receptors (AMPK, PDB ID: 2Y94 [55]; PPARα, PDB ID: 1K7L [56]) were used for computational analysis. Protein and ligand preparation was performed using UCSF Chimera software production version 1.14 [57]. Ligand compounds for soyasaponins were downloaded from the PubChem database [58] to create conformations using Marvin Sketch v17.1.30.

### 4.5. Molecular Docking

To estimate the binding affinity of soyasaponins to the target molecules, an in silico molecular docking study was performed. There are four docking programs used in this study to validate a better scoring option. Molecular docking for the set of optimized ligands was performed using the Auto Dock v.4.2.6 program [59]. The docking system looks for the most excellent introduction of the atoms to the dynamic location of each protein, using the Lamarckian genetic algorithm (LGA) [60]. Autodockvina employs an advanced angle-optimization strategy within the local optimization strategy. It is the stride form with more docking precision, counting using a modern scoring effort, and has productive optimization. Swiss Dock was based on the docking software EADock DSS [61]. The docking system generally calculates a score that is smaller than 0, indicating a higher binding affinity between a ligand and its receptor molecules [62]. Furthermore, Ledock (52) software was used because it had the best sampling power and 80.8% accuracy for the best conformations. The best score ligand compound for receptor binding was observed with UCSF Chimera V. 1.14. High-scoring ligands and control compounds were chosen for visualization. The 2D interaction of the protein–ligand complex structure, counting hydrogen bonds, hydrophobic interactions, and bond lengths, were analyzed using Ligplot+ [63] for high-affinity ties.

### 4.6. Molecular Dynamics Simulations

To perform the molecular dynamics (MD) simulations of the AMPK-SSC, SSB, or the AICAR (control) complex, and PPARα-SSC, SSB, or fenofibrate (control) complex structures, the Gromacs 5.1.2 package was utilized [64]. CHARMM36, which is an all-atom-drive lipid force field, was used to write the PPARα and AMPK [65,66,67] atomic drive field parameters. The topology of SSB, SSC, or the control atomic drive field parameters were obtained from the Gromos54a7 force field using the Automated Topology Builder (ATB, https://atb.uq.edu.au/index.py, accessed on 1 November 2021) [68], and the files were converted into the GROMACS file format. Initially, energy minimization was performed. After minimization, isobar isothermal ensembles (NPT) and canonical ensembles (NVT) were applied. The generation of MD runs was performed for 10 and 5 ns. The root mean square deviation (RMSD), root mean square fluctuation (RMSF), and binding energy were calculated after the runs. These parameters were outlined using the Gnuplot program [69].

### 4.7. Reagents and Antibodies

Soyasapogenol C was purchased from ChemFaces (Wuhan, China). Dimethyl sulfoxide (DMSO), sodium palmitate, compound C, and Oil Red O staining solution were obtained from Sigma-Aldrich (St. Louis, MO, USA). Protein concentration was measured using bicinchoninic acid (BCA; ThermoScientific, Waltham, MA, USA), and bovine serum albumin (BSA) was used as a standard. The PVDF membrane was obtained from Millipore Corp. (Billerica, MA, USA). The following primary antibodies were used for Western blot analysis: AMPK, p-AMPK (Thr172), TFIIB, and β-actin, which were purchased from Santa Cruz Biotechnology, Inc. (Dallas, TX, USA). PPARα was obtained from Abcam (Cambridge, UK).

### 4.8. Cell Culture and Treatment

HepG2 human liver cancer cells were purchased from the American Type Culture Collection (ATCC, Manassas, VA, USA). Cells were cultured in Dulbecco’s modified Eagle’s medium (DMEM) supplemented with 10% fetal bovine serum (Gibco, Carlsbad, CA, USA) at 37 °C in a humidified atmosphere of 5% CO_2_. The SSC was dissolved in 100% dimethyl sulfoxide (DMSO). The final concentration of DMSO did not exceed 0.1%.

### 4.9. Cell Viability

Cell viability was determined using the EZ-Cytox assay (DaeilLab, Seoul, Korea). HepG2 cells (5 × 10^3^/well) were seeded in 96-well plates and incubated with 0–20 µM SSC for 24 h. After treatment, the cells were incubated with the EZ-Cytox solution. The absorbance was read at 450 nm using a microplate reader (Tecan, Männedorf, Switzerland). The percentage inhibition due to SSC was obtained according to the formula: inhibition (%) = [(OD (sample) − OD (control))/(OD (normal) − OD (control))] × 100. All assays were performed in triplicate and then averaged.

### 4.10. Preparation and Treatment with Sodium Palmitate in HepG2 Cells

Palmitate was conjugated with fatty acid-free BSA (GenDEPOT, Barker, TX, USA) using a previously reported methodology [70]. Briefly, 69.6 mg of sodium palmitate was dissolved in 0.5 mL of 0.1 N sodium hydroxide at 70 °C to make a 500 mM stock solution. After the dissolution of palmitate, the stock solution was immediately added to serum-free DMEM (containing 5% fatty acid-free BSA) to obtain a 0.5 mM palmitate solution. The cells were pre-treated for 24 h with various concentrations of SSC (5–20 µM) before exposure to palmitate (0.5 mM) for 24 h.

### 4.11. Oil Red O Staining

HepG2 cells grown in 6-well plates were collected, washed with cold PBS, and fixed with 4% paraformaldehyde for 10 min. Cells were then incubated in 60% isopropanol for 5 min and stained with Oil Red O staining solution for 1 h. The cells were washed several times with ddH_2_O to remove excess stain. Images were captured using a microscope (Motic, CA, USA). 

### 4.12. Measurements of Intracellular Triglyceride Contents

Intracellular triglyceride (TG) content was measured using enzymatic colorimetric assay kits (Bio-Clinical System, Gyeonggi-do, Korea) after the lysis of the HepG2 cells with 1% Triton X-100 in PBS. Protein concentrations were determined using the BCA method with bovine serum albumin (BSA) as the standard. Intracellular triglyceride levels were normalized to cellular protein content.

### 4.13. Luciferase Assay

For luciferase assays, HepG2 cells were seeded at a density of 1 × 10^4^ cells in a 96-well plate. Cells were transfected with lipofectamine transfection reagent (Thermo Fisher Scientific, Rockford, IL, USA), and plasmids were used for transfection with the PPRE-X3-TK-LUC plasmid (Dr. Christoper K. Glass, University of California, San Diego, CA, USA) and PPARα expression vectors (Dr. Han Geuk Seo, Konkuk University, Seoul, South Korea). After transfection for 24 h, cells were treated with WY14643 (a PPARα agonist) [71] or SSC for 6 h. Luciferase activity was measured using the One-Glo Luciferase Reporter Assay System (Promega, Madison, WI, USA) and a luminescence plate reader (Berthold Technologies GmbH & Co., Bad Wildbad, Germany.

### 4.14. RNA Isolation and Real-Time Quantitative Polymerase Chain Reaction (RT-qPCR)

RNA from HepG2 cells was purified using the RiboEx Total RNA kit (GeneAll Biotechnology, Seoul, South Korea) according to the manufacturer’s instructions. Total RNA (2 µg) treated with ribonuclease (RNase)-free deoxyribonuclease (DNase) was reverse-transcribed with a Hyperscript^TM^ One-Step RT-PCR (GeneAll Biotechnology, Seoul, South Korea). RT-qPCR was performed using the SensiFAST^TM^ SYBR^®^ No-ROX kit (BIOLINE, UK) and a CFX Connect System (Bio-Rad Laboratories, Inc., Hercules, CA, USA). Relative gene expression levels were calculated using standard curve methodology, with GAPDH as an internal control. The experimental results obtained from qPCR were the Ct values. Three independent replicate tests were performed. The primer sequences used in this study are listed in Appendix A.

### 4.15. Western Blotting

Western blotting was performed as previously described in other studies [72]. Total cell lysates were boiled for 10 min with gel-loading buffer (0.125 M Tris-HCl, pH 6.8, 4% SDS, 10% 2-mercaptoethanol, and 0.2% bromophenol blue). Equal amounts of protein were separated using SDS-PAGE using 7–9% acrylamide gels, and then transferred onto polyvinylidene fluoride membranes (Millipore, Burlington, MA, USA) at 25 V for 10 min in a semi-dry transfer system (Bio-Rad Laboratories, Hercules, CA, USA). The membranes were immediately placed in blocking buffer (5% non-fat milk) in 10 mM Tris (pH 7.5), 100 mM NaCl, and 0.1% Tween-20. The blots were then blocked at room temperature for 1 h. The membranes were incubated with appropriate specific primary antibodies at 4 °C overnight, and then treated with horseradish peroxidase-conjugated anti-mouse and anti-rabbit antibodies (1:5000) at 25 °C for 1 h (Santa Cruz Biotechnology, Dallas, TX, USA). Protein bands were visualized using the SuperSignal^®^ West Pico Chemiluminescent Substrate kit (Advansta, San Jose, CA, USA) and Davinch-Chemi^TM^ (Davinch-K, Seoul, Korea).

### 4.16. Statistical Analyses

One-way analysis of variance (ANOVA) was used to determine the differences within treatments, and subsequently analyzed using the Bonferroni test in GraphPad Prism 5 (GraphPad, La Jolla, CA, USA) software. Statistical significance was determined using * *p* < 0.05 value. Data are expressed as the mean ± standard error of the mean (SEM) of three independent experiments.

## 5. Conclusions

In conclusion, the present study demonstrated that soyasapogenol C was more potent than soyasapogenol B in exerting anti-hepatic steatosis effects, using a novel SANDA methodology. The SANDA methodology is an effective and feasible method for screening potential candidates from natural products. In silico, soyasapogenol C had better ADMET characteristics and a higher binding affinity than soyasapogenol B, and the genes involved in lipid metabolism and metabolic syndrome associated with the disease were predicted using PPI network analysis. Soyasapogenol C stimulated phosphorylated AMPK and increased the nuclear translocation of PPARα in HepG2 cells in vitro. Furthermore, soyasapogenol C inhibited hepatic lipogenesis and increased fatty acid β-oxidation in HepG2 cells, thereby attenuating hepatic steatosis (Figure 8). This study provides a better understanding of the molecular actions of soyasapogenol C in exerting hepatoprotective effects, and it can be applied in the management of liver health in humans. Finally, SANDA could be a more effective methodology to identify potent compounds from dietary supplements, and the consumption of soyasapogenol C-rich fermented soybeans could prevent the development of hepatic steatosis-associated liver diseases. 

## Figures and Tables

**Figure 1 ijms-23-05468-f001:**
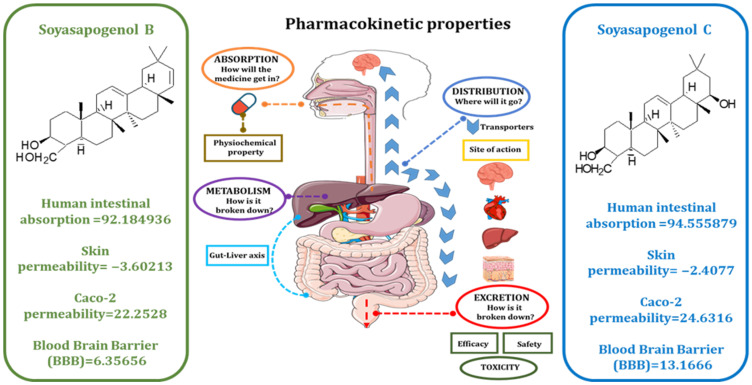
The comparative ADMET properties between soyasapogenol B and soyasapogenol C.

**Figure 2 ijms-23-05468-f002:**
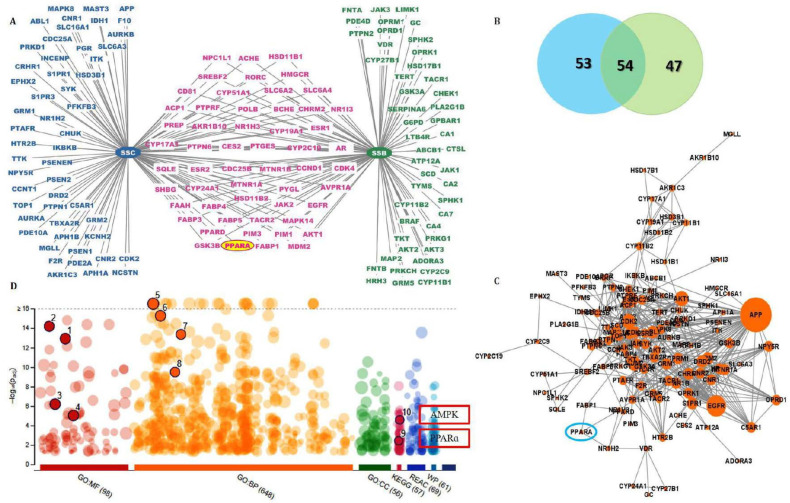
The network pharmacological analysis between soyasapogenol B and soyasapogenol C. (**A**) Target network prediction of SSB and SSC; (**B**) the Venn diagram of the targets of SSB–SSC; (**C**) Protein–protein interactions of the overlapped targets between SSB and SSC; (**D**) The enriched KEGG pathways of the overlapped targets between SSB-SSC using g: Profiler. Numbering represents 1—Kinase activity; 2—Protein kinase activity; 3—Lipid binding; 4—Kinase binding; 5—Lipid metabolic process; 6—Lipid biosynthetic process; 7—Regulation of lipid metabolic process; 8—Lipid catabolic process; 9—PPARα signaling pathway; 10—AMPK signaling pathway.

**Figure 3 ijms-23-05468-f003:**
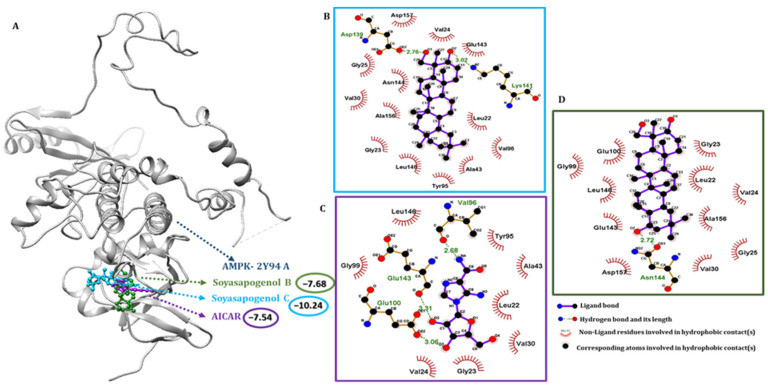
Molecular docking analysis using Auto Dock 4.2. (**A**) The binding interactions of soyasapogenol C (blue), soyasapogenol B (green), and the control compound AICAR (purple) with AMPK protein (grey); (**B**) 2D pharmacophore analysis between AMPK with active component soyasapogenol C (blue box); (**C**) 2D pharmacophore analysis between AMPK with control compound AICAR (purple box); (**D**) 2D pharmacophore analysis between AMPK with active component soyasapogenol B (green box).

**Figure 4 ijms-23-05468-f004:**
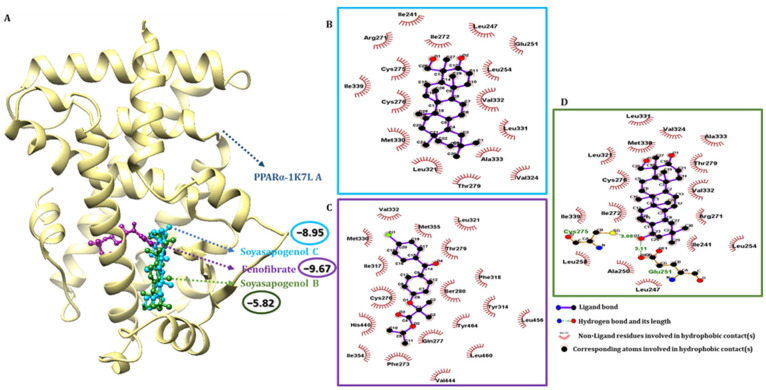
Molecular docking analysis using Auto Dock 4.2. (**A**) The binding interactions of soyasapogenol C (blue), soyasapogenol B (green) and fenofibrate (purple) with PPARα protein (gold); (**B**) 2D pharmacophore analysis between PPARα with active component soyasapogenol C (blue box); (**C**) 2D pharmacophore analysis between PPARα with control compound fenofibrate (purple box); (**D**) 2D pharmacophore analysis between PPARα with active component soyasapogenol B (green box).

**Figure 5 ijms-23-05468-f005:**
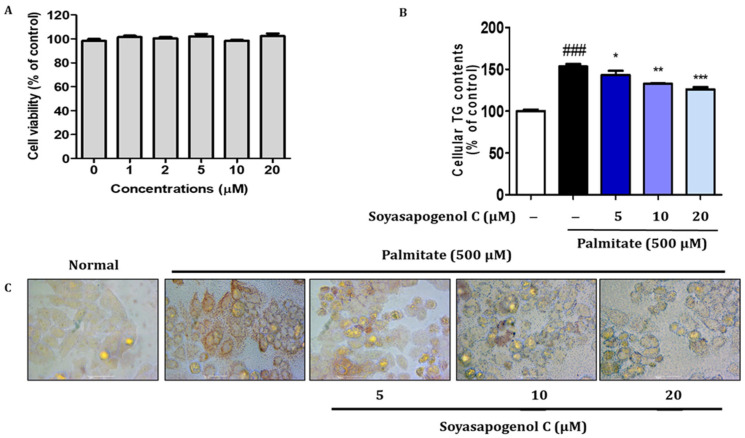
Effects of soyasapogenol C on lipid accumulation in palmitate-treated HepG2 cells. (**A**) HepG2 cells were treated with soyasapogenol C at various concentrations for 24 h, and cell viability was determined via Ez-Cytox assay. HepG2 cells were treated with palmitate for 24 h. (**B**) Measurement of intracellular triglyceride (TG) contents. (**C**) Lipid accumulation was determined via Oil Red O staining. Images of cells were photographed at 200× magnification. Data are presented as the mean ± SE of three independent experiments. ### *p* < 0.001 vs. blank, * *p* < 0.05, ** *p* < 0.01 and *** *p* < 0.001 vs. palmitate treated group.

**Figure 6 ijms-23-05468-f006:**
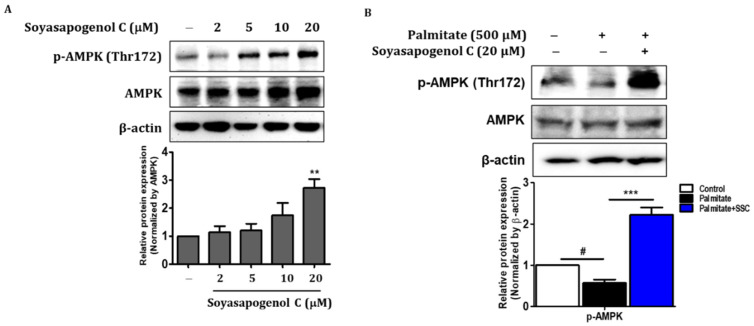
Effect of soyasapogenol C on AMPK activation in HepG2 cells. (**A**) Protein expression of phosphorylated AMPK was analyzed using Western blotting. (**B**) HepG2 cells were treated with palmitate (500 µM) and/or soyasapogenol C (5, 10, and 20 µM) for 24 h. Data are presented as the mean ± SEM of three independent expressions. ** *p* < 0.01 vs. normal; # *p* < 0.05, and *** *p* < 0.001 vs. palmitate-treated.

**Figure 7 ijms-23-05468-f007:**
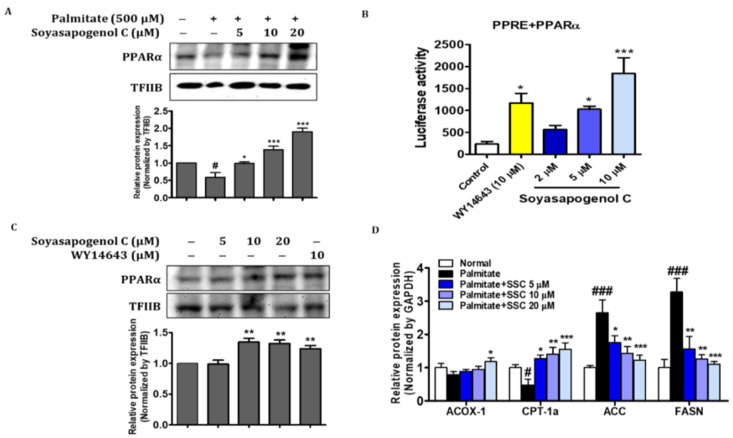
SSC increased transcriptional activity of PPARα and stimulated β-oxidation. (**A**) PPARα levels were analyzed in nucleus fraction using Western blotting. (**B**) For luciferase, the 3 × PPRE-TK-LUC plasmid and PPARα expression vector were transfected into HepG2 cells. Twenty-four hours after the transfection, the cells were treated with SSC or agonists (WY14643 10 µM) for 5 h. Values are expressed as the mean ± SEM of two independent replications. (**C**) PPARα levels were analyzed in nucleus fraction after treatment with SSC or agonists (WY14643 10 µM) for 24 h using Western blotting. (**D**) HepG2 cells were treated with palmitate (500 µM) and/or soyasapogenol C (5, 10, and 20 µM) for 24 h. Relative mRNA expression of ACOX-1, CPT-1α, ACC, and FASN. Data are presented as the mean ± SEM of three intendent expressions. # *p* < 0.05 and ### *p* < 0.001 vs. normal; * *p* < 0.05, ** *p* < 0.01, and *** *p* < 0.001 vs. palmitate-treated.

**Figure 8 ijms-23-05468-f008:**
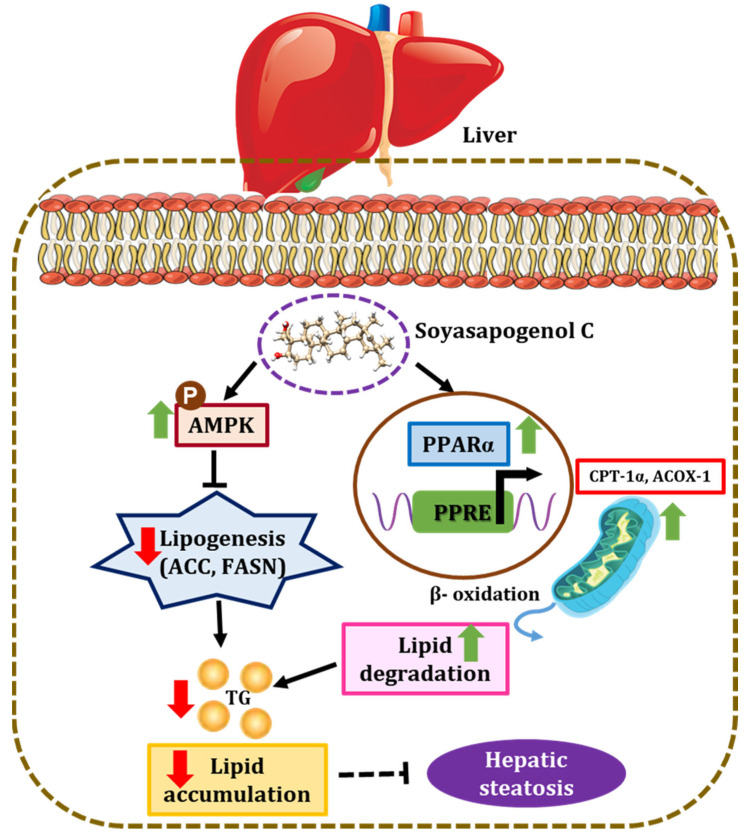
Schematic diagram representing the mechanism by which soyasapogenol C attenuates hepatic steatosis. Soyasapogenol C induced the activation of AMPK and increased the nuclear translocation of PPARα, which is associated with inhibition of hepatic lipogenesis (ACC and FASN) and stimulation of fatty acid β-oxidation (CPT-1α and ACOX-1) in HepG2 cells, thereby attenuating hepatic steatosis (↑ indicates upregulation; ↓ indicates downregulation).

**Table 1 ijms-23-05468-t001:** Detailed comparison of ADMET properties between control compounds of AMPK (AICAR) and PPARα (Fenofibrate) with soyasapogenol B and soyasapogenol C using the online tool preADMET.

Properties	AMPK (AICAR ^a^)	PPARα (FF ^b^)	SSB	SSC
**Absorption**				
Human intestinal absorption (HIA %)	18.27	97.39	92.18	94.56
Caco-2 cell permeability (nm s^−1^)	6.80	44.24	22.25	24.63
MDCK cell permeability (nm s^−1^)	0.58	15.527	0.044	0.048
Skin permeability (logKp, cm h^−1^)	−5.17	−1.55	−3.60	−2.41
**Distribution**				
Plasma protein binding (%)	5.12	100	100	100
Blood–brain barrier penetration (C_brain_/C_blood_)	0.63	0.11	6.36	13.17
**Metabolism**				
CYP2C19 inhibition	Non	Inhibitor	Non	Non
CYP2C19 Substrate	Non	Non	Non	Non
CYP2C9 inhibition	Non	Inhibitor	Inhibitor	Inhibitor
CYP2C9 Substrate	Non	Non	Non	Non
CYP2D6 inhibition	Non	Non	Non	Non
CYP2D6 Substrate	Non	Non	Non	Non
CYP3A4 inhibition	Inhibitor	Inhibitor	Inhibitor	Inhibitor
CYP3A4 Substrate	Weakly	Substrate	Substrate	Substrate
**Excretion**				
P-gp inhibition	Non	Inhibitor	Inhibitor	Inhibitor
**Toxicity**				
Ames test	Mutagen	Mutagen	Non	Non
Carcino_Mouse	Carcinogen	Carcinogen	Non-carcinogen	Non-carcinogen
Carcino_Rat	Non-carcinogen	Carcinogen	Non-carcinogen	Non-carcinogen

The color codes include green for highly positive/yes; yellow for weak positive; red for negative, no; blue for ADMET properties. ^a^ Acadesine (5-Aminoimidazole-4-carboxamide ribonucleotide); ^b^ Fenofibrate; The data were analyzed and obtained from (https://preadmet.qsarhub.com/, accessed on 22 July 2021) database.

## Data Availability

Not applicable.

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
