# Peer review of "Soyasapogenol C from Fermented Soybean (Glycine Max) Acting as a Novel AMPK/PPARα Dual Activator Ameliorates Hepatic Steatosis: A Novel SANDA Methodology"

_ijms, 2022, doi:10.3390/ijms23105468_

Round 1

Reviewer 1 Report

This is an interesting work by Arulkumar et al. who combined in silico screening with experimental work. It is methodologically well done and has important results.

I have only minor points to note:
A. Table 1: Reduce the number of decimal places (1-2). BBB penetration unit is missing. The information on metabolism is not helpful without values like IC50. Toxicity labelling: carcinogenicity labeling is confusing (positive/negative), use carcinogen or non-carcinogen - dosages would be necessary. Change title to better express that all the values in this table are estimates based on machine learning models, include the URLs of the tools used in the footnote.

B. Figure 2: Figure is partially not readable 

C. Page 6, line 167: "higher" instead of "high".

D. Several times the higher activity of SSC compared to SSB is mentioned: why was this not also shown in the experiments (Figure 6 and 7)?

Author Response

This is an interesting work by Arulkumar et al. who combined in silico screening with experimental work. It is methodologically well done and has important results.

I have only minor points to note A. Table 1: Reduce the number of decimal places (1-2). BBB penetration unit is missing. The information on metabolism is not helpful without values like IC50. Toxicity labelling: carcinogenicity labeling is confusing (positive/negative), use carcinogen or non-carcinogen - dosages would be necessary. Change title to better express that all the values in this table are estimates based on machine learning models, include the URLs of the tools used in the footnote.

Answer: Thank you for your suggestion and positive comments about the manuscript. We have changed the number of decimal places (1-2) in table 1. BBB penetration unit is Cbrain/Cblood ratio, added in the table. The tool is not providing any values or IC50 for metabolism. Toxicity labelling was corrected as carcinogen or non-carcinogen in the table 1. URL link was also added in the footnote.

B. Figure 2: Figure is partially not readable

Answer: We have replaced figure 2 with good resolution.

C. Page 6, line 167: "higher" instead of "high".

Answer: We have changed the text "higher" instead of "high" (Page 6, line 167, revised line 165)

D. Several times the higher activity of SSC compared to SSB is mentioned: why was this not also shown in the experiments (Figure 6 and 7)?

Answer: The study aims to screen potential compounds from fermented soybean (Glycine max) products by using the SANDA methodology.  In silico analysis revealed that SSB and SSC (Totally 15 distinct saponin compounds) both had drug-likeness properties but SSC had better ADMET characteristics and higher binding affinity with predicted targets. The main purpose is to choose biologically active compounds and evaluate the biological activities in vitro based on in silico analysis, which may help to reduce the experimental time while analyzing on a larger scale and is also very economical. We performed in vitro analysis for only SSC because SCC showed higher activity. This SANDA methodology will be an effective and feasible method for screening potential candidates from natural products.

Reviewer 2 Report

The paper by Radha Arulkumar Medina describes Soyasapogenol C (SSC) had high intestial absorption and binding interaction with AMPK and PPARα by a SANDA as well as high biological actibity to AMPK and PPARα in human hepatocyte. Furthermore, the authors showed SSC can be candidate for treatment of hepatic steatosis. The authors have some novel data but some of the methods need to be clarified

  1. The authors did not show any data SSC restore hepatic steatosis. They showed just SSC reduce lipid accumulation and upregulate AMPK and PPARα activation in HepG2. (title, result and discussion)
  2. Do PPARα and AMPK activations are independently effected by SSC? If SSC indirectly effect on PPARα activation via AMPK, the authors are not able to describe SSC is a dual activator of PPARα and AMPK. (title)
  3. SSC is not lowest intermolecular energy in some docking tool. Line 183-185 is unclear.
  4. Table S5 is primer list. Please make sure what the table was indicated. (line 220)
  5. Please correct finale figure number. It may be figure 7.

Author Response

The paper by Radha Arulkumar Medina describes Soyasapogenol C (SSC) had high intestinal absorption and binding interaction with AMPK and PPARα by a SANDA as well as high biological activity to AMPK and PPARα in human hepatocyte. Furthermore, the authors showed SSC can be candidate for treatment of hepatic steatosis. The authors have some novel data but some of the methods need to be clarified

The authors did not show any data SSC restore hepatic steatosis. They showed just SSC reduce lipid accumulation and upregulate AMPK and PPARα activation in HepG2. (Title, result and discussion)

Answer: Thank you for your suggestion and positive comments about the manuscript. Hepatic steatosis is defined as the accumulation of lipids in hepatocytes [1]. We demonstrated that SSC ameliorates hepatic steatosis in HepG2 cells, not restore hepatic steatosis. SSC significantly inhibited triglyceride accumulation, which was associated with the suppression of lipogenesis genes and enhancement of fatty acid oxidation gene expression in HepG2 cells.

Do PPARα and AMPK activations are independently affected by SSC? If SSC indirectly effect on PPARα activation via AMPK, the authors are not able to describe SSC is a dual activator of PPARα and AMPK. (title)

Answer: Thank you for your valuable suggestion. of SSC had a higher affinity with AMPK and PPARα proteins (Figure 3 and 4, respectively). The molecular dynamics (MD) simulation technique helps to mimic the conformational changes of a protein-ligand system over time. MD simulation results suggest that SSC had lower fluctuations when compared to the control within 5 nanoseconds for both AMPK and PPARÉ‘ (Fig. S2 and S3, respectively). The interaction energy value indicates the active interaction of AMPK-SSC and PPARα-SSC, resulting in the formation of a stable complex. Further, SSC induced the phosphorylation of AMPK and stimulated nuclear translocation of PPARÉ‘ in HepG2 cells. Also, SSC enhanced PPAR response element (PPRE) binding activity in HepG2 cells. Both in silico and in vitro results suggest that SSC independently activates AMPK and PPARÉ‘ and it could be a dual activator of AMPK/PPARα.

SSC is not lowest intermolecular energy in some docking tool. Line 183-185 is unclear.

Answer: Thank you for your valuable suggestion. We have revised the lines 183-185 according to the results.

Lines 183-185 (revised line 182-184): SSC had a higher binding interaction for AMPK and PPARα in at least two docking tool scores (Table S3) and also found high binding affinity and lowest intermolecular energy in comparison to the control (Table S4).

Table S5 is primer list. Please make sure what the table was indicated. (Line 220)

Answer: We have corrected Table S5 to Table S4 (Lines 217 and 220, revised line 216 and 218, page 07), according to your suggestion.

Please correct finale figure number. It may be figure 7.

Answer: We have corrected the final figure number 7 (Line 335, revised line 333, page 11), according to your suggestion.

References

  1. Nassir, F.; Rector, R. S.; Hammoud, G. M.; Ibdah, J. A., Pathogenesis and Prevention of Hepatic Steatosis. Gastroenterol Hepatol (N Y) 2015, 11, (3), 167-75.

Round 2

Reviewer 2 Report

The authors have made substantial revisions in response to the original critique, and the manuscript is greatly improved as a result. However, there are still unclear points.

  1. It is still unclear revised line 182-184. As the authors described, SSC had a higher binding interaction for AMPK in at least two docking tool scores (Table S3) but no docking tools did not show SSC had a higher binding interaction for PPARa in comparison to the control. Moreover SSC had binding affinity for PPARa is lower and intermolecular energy for PPARa is higher in comparison to the control (Table S4). Please clarify the sentences.

Author Response

The authors have made substantial revisions in response to the original critique, and the manuscript is greatly improved as a result. However, there are still unclear points.

It is still unclear revised line 182-184. As the authors described, SSC had a higher binding interaction for AMPK in at least two docking tool scores (Table S3) but no docking tools did not show SSC had a higher binding interaction for PPARa in comparison to the control. Moreover, SSC had binding affinity for PPARa is lower and intermolecular energy for PPARa is higher in comparison to the control (Table S4). Please clarify the sentences.

 Answer: Thank you for your suggestion and positive comments about the manuscript. We have revised the lines 182-184 according to the results.

Revised lines 182-187: SSC had a higher binding interaction for AMPK in at least two docking tool scores than control and SSB (Table S3). Likewise, SSC had a higher binding score for PPARα in at least two docking tool scores than SSB which is almost similar to control. Furthermore, SSC had higher binding energy and lowest intermolecular energy for AMPK when compared to the control, and a good score for PPARα (Table S4).